# The Secondary Use of Data to Support Medication Safety in the Hospital Setting: A Systematic Review and Narrative Synthesis

**DOI:** 10.3390/pharmacy9040198

**Published:** 2021-12-13

**Authors:** Navila Talib Chaudhry, Bryony Dean Franklin, Salmaan Mohammed, Jonathan Benn

**Affiliations:** 1Research Department of Practice and Policy, UCL School of Pharmacy, 29–39 Brunswick Square, London WC1N 1AX, UK; bryony.franklin@nhs.net; 2Centre for Medication Safety and Service Quality, Pharmacy Department, Imperial College Healthcare NHS Trust, London W6 8RF, UK; 3NIHR Patient Safety Translational Research Centre, Department of Surgery and Cancer, Faculty of Medicine, St Mary’s Campus, Imperial College London, 5th Floor Wright Fleming Building, Norfolk Place, London W2 1PG, UK; s.mohammed13@imperial.ac.uk; 4NIHR Yorkshire and Humber Patient Safety Translational Research Centre, School of Psychology, University of Leeds, Woodhouse Lane, Leeds LS2 9JT, UK; J.Benn2@leeds.ac.uk

**Keywords:** electronic systems, medication safety, secondary use of data, quality improvement, hospital

## Abstract

Objectives: To conduct a systematic review and narrative synthesis of interventions based on secondary use of data (SUD) from electronic prescribing (EP) and electronic hospital pharmacy (EHP) systems and their effectiveness in secondary care, and to identify factors influencing SUD. Method: The search strategy had four facets: 1. Electronic databases, 2. Medication safety, 3. Hospitals and quality/safety, and 4. SUD. Searches were conducted within EMBASE, Medline, CINAHL, and International Pharmaceutical Abstracts. Empirical SUD intervention studies that aimed to improve medication safety and/or quality, and any studies providing insight into factors affecting SUD were included. Results: We identified nine quantitative studies of SUD interventions and five qualitative studies. SUD interventions were complex and fell into four categories, with ‘provision of feedback’ the most common. While heterogeneous, the majority of quantitative studies reported positive findings in improving medication safety but little detail was provided on the interventions implemented. The five qualitative studies collectively provide an overview of the SUD process, which typically comprised nine steps from data identification to analysis. Factors influencing the SUD process were electronic systems implementation and level of functionality, knowledge and skills of SUD users, organisational context, and policies around data reuse and security. Discussion and Conclusion: The majority of the SUD interventions were successful in improving medication safety, however, what contributes to this success needs further exploration. From synthesis of research evidence in this review, an integrative framework was developed to describe the processes, mechanisms, and barriers for effective SUD.

## 1. Introduction

Hospitals have a variety of data available within electronic prescribing (EP) and electronic hospital pharmacy (EHP) systems. These data have considerable potential to be re-used to improve patient care and drive quality improvement [1]. It is not known whether the data held within EP and EHP systems are currently being used effectively to support improvement in medication quality and safety, or the factors affecting its use.

‘Secondary use of data’ (SUD) has been defined as ‘reuse of clinical and/or operational data for purposes other than direct patient care’ [2]. Previous research has suggested that SUD could drive improvement within healthcare, preventing reinvention and/or duplication of data for clinical research purposes while reducing resource costs, and potentially facilitating sustainability of improvement as the data are readily available [3,4,5]. For example, quick and effective electronic extraction of data supports feedback to healthcare providers that is effective, timely, and continuous [6]. Bradley et al. suggest that feedback is most effective when: 1. data are perceived to be valid by receivers, 2. the credibility of the data is established, 3. the source and timeliness of data is clear to receivers, 4. the feedback provides performance benchmarking, 5. feedback is provided by leaders, 6. feedback is personalised and 7. sustained over time [1].

Case studies demonstrate how healthcare organisations have developed pseudonymised or anonymised databases to monitor particular outcomes [7,8]. A previous review of SUD from electronic health record (EHR) systems focused on quality of coding, information quality, and use of natural language processing as the principal mechanisms for reuse [9]. Other reviews have focused on data use and its effectiveness for monitoring healthcare associated infections, illustrating the use of medico-administrative or clinical and laboratory-based data for electronic surveillance and demonstrating that systems are generally sensitive in acquiring information, but the information they retrieve is not always specific to requirements [10]. Data surveillance has been used to improve quality in relation to omitted doses and to measure adherence to prescribing guidelines [11,12]. 

In the reduction of medication errors, electronic trigger tools, smart pumps, and bar code technology have been employed in many hospitals [13,14,15]. Previous systematic reviews have explored the use of data from these electronic systems [16,17,18], however, there have been no reviews specifically focusing on SUD from EP and EHP systems. Use of existing EP and EHP data could help healthcare professionals reflect on their performance, improve services, and encourage quality improvement, but it is not yet known how SUD can be effectively applied to achieve this. More specifically, little is known about existing SUD interventions and their effects, including how SUD is defined by researchers working in this area, what types of systems and data are utilised in practice for this purpose and by whom. Our objectives were to: (1) identify the types of interventions that aim to improve quality and/or safety of medication use that are based on SUD from hospital EP and EHP systems, (2) synthesise the available evidence for the efficacy of these interventions, and (3) identify the factors influencing the SUD process. 

## 2. Materials and Methods

A systematic literature review was conducted, with narrative synthesis selected due to the anticipated heterogeneity of the literature [19,20]. PRISMA reporting guidelines were followed in combination with Realist and Meta-narrative Evidence Syntheses: Evolving Standards (RAMESES) reporting guidelines [21,22].

### 2.1. Search Strategy

Two search strategies were formulated and structured around four main facets reflecting the PICO format [23]. The first search strategy had three facets (‘electronic systems’, ‘medication safety’, and either ‘hospital’ or ‘quality and safety’) and was designed to maximise sensitivity in identifying SUD interventions (Table 1); the second had one facet (‘secondary use of data’) to emphasise specificity in identifying literature that focused on factors affecting SUD. Appendix A, respectively) presents a detailed justification for each facet together with the Boolean terms used. 

### 2.2. Definitions

The following definitions were adopted:‘Secondary use of electronic prescribing/pharmacy data’: the reuse of clinical and/or operational data from an EP or EHP system for purposes other than direct patient care or the original purpose for which the data were used.‘Intervention based on secondary use of electronic prescribing/pharmacy data’ (‘SUD intervention’): The reuse of the data from an EP or EHP system for secondary purpose(s) with the intent of changing or improving a process, either alone or in combination with other intervention(s). The actual implementation of an electronic system was not considered an intervention in this context.

### 2.3. Study Selection

Papers were included if they were empirical studies that described and/or evaluated an intervention based on SUD from hospital EP and/or EHP systems with the intention of improving outcomes in relation to the safety and/or quality of medication use (i.e., classed as intervention studies and restricted to hospitals). Papers presenting factors to consider for SUD that described more than one stage for reusing data in healthcare (not restricted to hospitals) were also included and classed as non-interventional literature; empirical studies and case studies were included but editorials and commentaries excluded. The non-interventional literature was not restricted to hospitals as factors affecting secondary use of data and lessons learnt could be applied across different settings; whereas the interventional studies were restricted to hospitals only as we were interested in the secondary use of EP and EHP data. 

Studies that used data from the following were excluded: electronic trigger tools, barcode technology, smart pumps, unit dose cart exchanges, automated medication storage systems, incident reporting systems, post-marketing drug surveillance, laboratory databases, standalone clinical decision support systems (defined as ‘software that is designed to be a direct aid to clinical decision-making’) [24], and paper-based prescribing. If a paper omitted key information that would determine its inclusion or exclusion, study authors were contacted to request the information concerned. 

### 2.4. Screening Process, Data Extraction, and Analysis

NTC conducted the title, abstract and full text screening with a second check conducted at each stage by a second researcher SM (10% titles and abstracts, approximately 50% full text record screening for intervention studies, and 10% for non-interventional studies). During full text screening, if a paper cited use of an electronic system to obtain medication-related data from EP and/or EHP but did not specify which type of system, the authors were contacted for more information, but the paper was still included even if no response was received. Disagreements between NTC and SM during full text screening were reviewed by BDF and JB, and a consensus on inclusion reached following discussion. Cohen’s Kappa was calculated to determine the consensus between the two reviewers. Data were extracted from all included studies by NTC using a piloted data extraction form, and then checked by SM. Based on the papers included, a descriptive analysis was conducted with inductive analysis to produce high level themes. Principles for effective feedback [1] were used to theoretically inform analysis of included articles, to determine whether these principles were used for the feedback interventions for the interventional studies included. The literature obtained was methodologically heterogeneous and therefore we were unable to conduct formal quality assessment. All references of included articles were screened to snowball further relevant articles to capture a broader range of literature. Any descriptive information regarding factors influencing SUD in the articles included was used to synthesise a conceptual framework to provide a holistic overview of the different factors in the current literature. The protocol for this review was registered with PROSPERO international prospective register of systematic reviews (registration number: CRD42016042925).

## 3. Results

### 3.1. Overview of Included Literature

After electronic and manual de-duplication, 5476 articles remained for title screening. Cohen’s Kappa values for inter-rater reliability of title and abstract screening were 0.8 (substantial agreement) and 0.6 (moderate agreement), respectively. Full text screening was conducted for 355 articles, of which 14 were subsequently included and 341 excluded (see Figure 1 for PRISMA diagram). 

In total, nine studies were classed as interventional studies and five as non-interventional. Of the nine interventional studies (Table 2), six were from the UK, and three from the USA. Of the six UK studies, four used data from EP systems [11,25,26,27], one from a EHP system [28], and one did not specify which [29]. Of the three US studies, one used data from an EP system [30], one from an EHP system, [31] and one from both [32]. Of the nine interventional studies, six were quantitative evaluations of interventions based on SUD, of which one was in a large acute hospital using a retrospective time series analysis design [25], one in a tertiary hospital using a controlled before-and-after design [32], one in a general district hospital using an uncontrolled before-and-after design [28]. 

Three did not specify hospital types, [26,27,30] one of which used a time series design [27], the second used an uncontrolled before-and-after design [30], and the third used retrospective data analysis [26]. Of the remaining three interventional studies, two were ethnographic SUD evaluations [11,29], and one was a qualitative intervention study [31]. Of the five non-interventional studies (Table 3), one was a qualitative paper exploring expert views [33], one was a reflective report of a case study [34], and three were literature reviews (one narrative [3] and two systematic [35,36]) Of these five papers, two were from USA [33,34], and one each from Denmark [35], Austria [36], and the UK [3]. 

### 3.2. Definitions of SUD Employed in the Literature

Only two of the nine interventional studies specifically mentioned the term ‘secondary use of data’ [11,29]. One of these described SUD as ‘data amassed in the databases of electronic patient records and other clinically-oriented information systems…’ that ‘…can be a valuable resource for purposes other than direct clinical care’ [11]. This implies that SUD refers to aggregated data, as supported by all nine included studies [11,25,26,27,28,29,30,31,32]. Another paper defined the primary purpose of EHR as ‘recording information about the care of individual patients’, suggesting that anything beyond this be considered as secondary use [11]. Of the five non-interventional papers, one defined SUD as ‘non-direct care use of personal health information including […] quality/safety measurement […] including strictly commercial activities’ [33]. For three of the five papers, definitions of SUD were implicit in the descriptions of the interventions, including use of routinely collected data [3], data exchanged or reused for clinical or non-clinical purposes [35], and reuse of existing health information for secondary purposes (for example medical research, decision making in health policies, and quality assurance in healthcare) [36].

**Table 3 pharmacy-09-00198-t003:** Five non-interventional studies looking at factors affecting secondary use of data.

Author(s)	Title	Country	Type of Study:	Factors to Consider When Reusing Data:
Bain et al., 1997 [3]	Routinely collected data in national and regional databases—an under-used resource	UK	Narrative literature review	Consists of 5 main stages to consider: Identification of useful data routinely entered; Confidentiality/privacy/ethical issues; Data quality (data completeness, data accuracy and validity, timeliness of data); Appropriateness of the routine data for the selected purpose; Analysis and interpretation of the data.
Danciu et al., 2014 [34]	Secondary use of clinical data: The Vanderbilt approach	USA	Reflection on a case study	9 factors to consider: The need of clinical enterprise software to undergo data extraction; Data identification: this can be done using different codes; De-identification of data and storage: removal of personal information; Specialist skills and knowledge: required to keep repository updated; Presenting the need for this data being reused; Improving systems due to incentive of gaining access to reusing the database; Translational use of clinical data envisaged and supported; Making access to data warehouse known and available to all at all levels; Having a request process infrastructure that supports secondary use of data.
Galster, 2012 [35]	Why is clinical information not reused?	Denmark	Systematic literature review	4 major barriers identified resulting in lack of clinical data reuse: Lack of data availability when required; The data source use is prohibited; Data cannot be used in the form available (reasons for this spilt in three categories technical, political, and quality reasons); Data are inadequate to be reused (two main reasons insufficient reliability and inadequate relevance). All placed under technical, organisational, legal, and medical issues.
Holzer and Gall, 2011 [36]	Utilizing IHE-based [Integrating the Healthcare Enterprise] Electronic Health Record Systems for Secondary use	Austria	Systematic literature review	Requirements for secondary use of data from electronic health records: Factors to consider for secondary use of data: Security measures to be considered, data formatting, user groups for secondary use of data and their requirements and query formulation process. System requirements (eight in total): standard terminology, cross patient/domain retrieval, selection of document, anonymisation, query within retrieved document, user roles, compliance with secondary use of data policies, and sensitisation within population.
Safran et al., 2007 [33]	Toward a national framework for the secondary use of health data	USA	Qualitative work—discussion between experts.	5 recommendations: Increase transparency of data use and promote public awareness, focus ongoing discussions on data access, use, and control not on ownership; Discuss privacy policies and security for secondary use of health data increase public awareness of benefits and challenges associated with secondary use of health data; Create taxonomy for secondary uses of health data; Address comprehensively the difficult, evolving questions related to secondary use of health data and focus national, and State attention on the secondary use of health data.

### 3.3. Interventions Based on SUD in Hospitals (Interventional Studies)

#### 3.3.1. Types of EP Systems and the Data Used

In total, seven of the nine interventional studies reused EP data, of which two did not specify the EP system used [26,30]. The remaining five used Health Evaluation through Logical processing (HELP) [32], Electronic Prescribing and Decision Support System (ePDSS) [11], Prescribing Information and Communication System (PICs) [25,27], and Innovian, Carevue, Metavision, and QS [29]. Only three used EHP systems, of which one was linked to an EP system [32] and two were not specified [28,31]. 

The data extracted included class of medication and information about the medication (i.e., dose, interval, route of administration, overdue doses, errors, missed medication, time lapse between medication being prescribed and administration of first dose, quantity of drugs dispensed, prescribing trends, and overview of medication administration) [11,25,26,27,28,29,30,31,32].

#### 3.3.2. Types of Intervention Based on Secondary Use of Data

The aims of the nine interventional studies were to reduce medication administration errors such as missed doses [11,25,27,32], improve prescribing (focusing on prescribing errors, adherence to guidelines, and inappropriate prescribing) [26,28,29,30,31], and make process improvements related to medication safety [29]. Electronic data were consistently used in conjunction with other interventions, forming complex interventions in all nine studies [37]. In total, four categories of SUD intervention were identified, with some studies including more than one category of intervention: feedback informed by SUD to healthcare professionals, [11,25,27,28,29,30,31,32], incorporation of additional features into an EP system (i.e., addition of a pause function [25] or upper dosing limits [26]), production of prescribing guidelines [28], and educational interventions on errors [32]. SUD was used to provide feedback in eight papers, [11,25,27,28,29,30,31,32] via dashboards [25,27], in person [11,30], posters [32], emails [11,25,30], visual indicators [25], care omission/audit meetings [11,25,27,28], and reports [31]. 

#### 3.3.3. Secondary Users of Electronic Data

Individuals receiving the data as a form of SUD intervention were physicians [27,30,31], nurses [25,30,32], and managers [11,27]; other papers mentioned ‘members of staff’ or ‘frontline staff’ without specifying their profession or job role [11,28]. The teams involved in implementing and/or evaluating these interventions were stated in four studies: physicians [26,30], pharmacists [28,30], researchers [30], and the informatics department [25]. 

#### 3.3.4. Effectiveness of Secondary Use Interventions

Of the nine interventional studies (Table 2), six quantitatively evaluated SUD interventions. Three reported success in improving medication-related outcomes, including feedback and education sessions that increased real-time charting and timely documentation of medication administration [32]; production of a new policy that significantly reduced prescribing and use of temazepam [28] and clinical dashboards and meetings were associated with significant reductions in both missed doses and mortality rates [27]. Of the remaining three, one quantitative study reported success in reusing data to increase sensitivity and specificity of dose limits applied to various medications [26]. The remaining two quantitative studies reported multiple SUD interventions with mixed success in improving the targeted medication safety outcomes [25,30]. Of the two quantitative studies, one implemented four interventions, three of which resulted in reduction in missed antibiotic and non-antibiotic doses. These three interventions were introduction of clinical dashboards, pausing e-prescriptions, and executive-led overdue doses root cause analysis (RCA) meetings [25]. The fourth intervention, a visual indicator for overdue doses, did not result in a significant decrease in missed antibiotics or non-antibiotic doses [25]. In the second quantitative study, feedback was provided via email and resulted in reduction of narcotic prescribing errors by 83% with the number of days between successive narcotic prescribing errors increasing, whereas the antibiotic prescribing error rate remained the same [30]. 

All three of the non-quantitative interventional studies qualitatively explored whether the interventions were perceived to be successful [11,29,31]. Two of the three studies suggested that multiple interventions were needed for SUD to be successful [11,31]; the third study implied that users were able to successfully reuse data for its primary and secondary purposes but faced challenges when reusing data for secondary purposes [29]. The challenges were caused by lack of primary data entry, missing information, and cross checking information due to poor data quality [29]. 

The use of regular feedback, having strategic goals for quality improvement and board/senior level involvement was perceived to have helped achieve objectives relating to SUD interventions [11,25]. It was perceived as being important to build consensus among clinical leadership [31] and other staff to facilitate bidirectional communication while answering queries such as the purpose for SUD [30]; distribution of work was also deemed important to produce a positive outcome, including having experienced clinical staff with knowledge of information technology (IT) to bridge the gap between IT and clinical practice [29]. Other initiatives to enhance SUD included automating data entry, providing smart forms, integrating data into workflow, prioritizing data entry, increasing awareness of data usage, structuring free text data, supporting manual extraction, and supplying visual representations of data [29]. It was also deemed important to ensure information was timely, formatted, and valuable to the recipients [30]. 

SUD was perceived to help generate intelligence, and to make practices, behaviours, and performance visible as well as the actions necessary as a result [11]. SUD enabled quick real-time data production [11], personalised timely feedback [30] and initiation of additional improvement [11]. SUD could not be implemented alone, however, and required additional activities to achieve the intended outcome [11]. 

Some of the unintended effects of SUD included individuals becoming more aware that their behaviour was being monitored, leading them to focus more on electronic documentation compared to non-electronic tasks, and stress prior to RCA meetings [11]. SUD was often not considered during system implementation, resulting in additional work to repurpose data. This included having to deal with high volumes of unstructured data, improve data entry [29] due to poor data quality [31], and additional fields being required to help overcome data limitations and capture additional information [31]. In order to support such changes, additional finance was needed to support SUD as a sustainable, continuous process [29]. 

#### 3.3.5. Limitations of Included Studies

In two studies, exclusions resulted in potential limitations, e.g., exclusion of specific patient groups/services precluding comparison among groups [25,27], and not including unrecorded doses as omitted doses [25]. Practical limitations were experienced in one study where large datasets could not be analysed; this could have provided an insight into organisational factors affecting the study outcome [25]. In two studies, two interventions were implemented close together, precluding assessment of the individual impact of each [25,32]. The main limitation encountered within five studies was lack of detail of the intervention, outcome measures and analysis, e.g., of these include: no detail on the individual delivering feedback [32], lack of exploration around maintenance of outcomes 1 year post intervention [32], lack of narrative around policy implementation [28], diagnosis and survival rates of patients not included [27], lack of qualitative data presented from users perspective from group meetings or feedback received [31], lack of user perspective on data acceptance [28] and no descriptive analysis [26]. There were study design limitations in four studies, specifically movement of staff between intervention and control units [32], inability to determine sole impact of one intervention due to multiple interventions being implemented at the same time in a time series analysis [25], inconsistent interview styles used between senior members of the hospital and frontline staff members in a single case study [11], and no additional data collection in a qualitative study to assess success of SUD beyond qualitative interviews and observations that could have been subject to the Hawthorne effect [29]. 

### 3.4. Factors Influencing Secondary Use of Data in Healthcare (Interventional and Non-Interventional Studies)

The mechanisms reported to influence the efficacy of SUD interventions included: use of regular feedback [11,25], involvement of clinical leadership [31] and senior members of staff [11,25], bidirectional communication [30], having staff with IT and clinical knowledge [29], high data quality [31] and additional finance to support SUD interventions [29]. The main advantages and disadvantages of SUD were data quality (completeness and volume of data available), timeliness of and the process of reusing data, and the perception of individuals receiving the data (i.e., acceptability). The barriers identified in one study were lack of data availability, data access limitations and inadequate data (i.e., reliability and relevance of the data) [35]. The requirements for effective SUD included (1) improving systems for data reuse, support and awareness of SUD, and (2) presence of some form of data warehouse [34,35,36]. 

From the included literature, it was possible to identify distinct steps reported within descriptions of the process of reuse of data. Nine general steps were identified in total: identifying data [3,34], defining outcome measures, [11,25,27,28,31] achieving clarity around specific tasks assigned to individuals within the team reusing data [11], considering confidentiality and ethical issues [3,34], determining data quality [3,25], consideration of data being appropriate [3], data linkage [31], data extraction [25,27,31,34], and data analysis [3,25,34]. From the interventional and non-interventional studies included, four main factors were identified that may influence these different steps: the systems being used (i.e., the human and system interaction and data quality) [25,29,32,33,34,35,36], the organisation (i.e., context) [33,34,35], users of secondary data [25,30,32,34,36], and privacy policies and security [33]. 

A conceptual framework was synthesised from the data extracted from included studies regarding the reported processes of SUD and the factors, both positive and negative, that influence its effectiveness (Figure 2). The advantages of SUD fell within two themes: the system—providing good data quality and large volumes of detailed data, [25,32], and the SUD process—perceived to be a quick, an easy form of reporting, reducing bias, and facilitating easy monitoring of outcomes [11,25,30]. The main barriers were technological and organisational: inadequate data (i.e., not reliable or not relevant), lack of data available for reuse, and restricted access to data sources [35]. The reported disadvantages of SUD focused around three themes: the process of SUD [25,29,30], the system (data quality) [11,25,29], and users (the knowledge of the person using the data and selected method of data delivery) [25,30,32]. The disadvantages around the SUD process identified were difficulty in reusing data straight after electronic system implementation, and difficulty in determining the value and effect of SUD as interventions can be complicated [25,29,30]. 

## 4. Discussion

### 4.1. Summary of Main Findings

This review provides an overview of SUD for quality and safety of medication use, through systematic description and synthesis of the limited evidence base within this area. Our analysis explores the definitions of SUD used in current research and describes the context and key features of reported SUD interventions, including the types of EP/EHP systems used, the types of data used and the design of the SUD intervention itself, including the secondary users of the data. In summarising the evaluative evidence for secondary use interventions, we identified nine interventional studies presenting four different types of SUD interventions, with feedback the most common. Other types of SUD intervention were incorporation of additional features into EP systems, production of guidelines, and education. In the multi-interventional studies included in this review, only two reported not improving the targeted medication safety outcomes; these were feedback via visual indicators which did not have a significant effect in reducing omitted doses [25] and feedback via email, which had no significant impact on reducing antibiotic prescribing errors [30]. Four main factors were found to influence SUD in the research literature: the organisation (relating to support and data accessibility), the system (data quality), the users of SUD (knowledge), and policies and security (data privacy). In synthesising the findings and lessons learnt from existing research in this area, it was possible to construct an integrative framework to promote consistency in definitions and support design and implementation of effective SUD systems and processes (Figure 2). 

In considering reporting of secondary users in the included studies, there was generally a lack of consideration of the views of those receiving the data. Data sources and timeliness were generally stated but it was not clear whether concerns around the data quality, validity, or appropriateness were explored from the perspective of individuals receiving the data. 

### 4.2. Comparison to Previous Literature

Previous systematic reviews have explored selected aspects of SUD, e.g., re-identification of personal information, completeness of data, and quality assessment [38,39,40]. However, none have provided an overview of the steps around SUD and the factors that may influence the process aiming to enhance the quality and/or safety of medication use. Two reviews concluded there was limited evidence around the reuse of data [9,41] and the present review has therefore helped elaborate on this. There has been some information regarding the recipients receiving data for secondary purposes but there is lack of reporting around the wider impact of interventions, the sociotechnical elements and social context, similar to the findings of this review [41]. 

The key three factors identified in our review that influence SUD resemble the framework produced by Cornford and colleagues to evaluate the efficiency, utility, and impact of an electronic system for tropical diseases [42], i.e., the organisation (i.e., context), the user (i.e., human perspective), and the system [42]. However, this review has added another element which is policies and clarification on what the process of secondary use of data is upon which these factors have an impact. 

A more recent review focusing specifically on SUD for antimicrobials identified two studies of SUD interventions; this concluded that data from EP and/or EHP systems could be used for evaluating or supporting antimicrobial stewardships, and similar to the present review’s findings, that better system functionality is needed [43]. However, the present review explores SUD for a broader range of medication types rather than being restricted to antimicrobials, and provides a framework presenting the process of SUD and the factors affecting it. Therefore, healthcare providers could review these factors in their organisations before reusing existing electronic data to help optimize the outcomes they wish to achieve and reap the benefits of reusing existing data. 

### 4.3. Implications for Practice

When designing SUD systems and processes, our review of the current evidence suggests that taxonomies should be produced for SUD [33], with standardised terminology and data formatting [36], and a focus on better quality data [34]. Organisational factors that would enhance SUD include better organisational support [34]; raised awareness around SUD [34]; enhanced data access [35]; provision of platforms similar to data warehouses [34]; and enhanced transparency of data [33]. From the user perspective, having the right skills and knowledge, having a clear purpose [34], understanding the data and how to use it [25], and being aware of potential resistance to feedback [30] may be important factors. Future developments need to address concerns reported in the empirical literature concerning policies relating to privacy and security of data for secondary use [33]. Further research and development is required to realise the potential of routinely available data for medication safety, increase the return on investment by healthcare providers in making SUD and including better linkage to systems that can facilitate and optimise secondary use, such as decision support systems.

This review has highlighted a range of shortcomings with existing systems and processes for SUD in medication use, linked to five principal categories: (1) Organisation (stakeholder engagement and managerial support, and access to data and resources (software and hardware)), (2) Technology (system functionality to extract data, and improved data quality (based on primary use of system)), (3) Users (knowledge and awareness of data that exists, knowledge of data analysis and interpretation (in relation to the context extracted and to be applied within), knowledge of the audience presenting the data too, and knowledge of data timeliness, accuracy, validity, and completeness), (4) Policy (the security and data protection need to be considered when reusing data, i.e., potential risk of re-identification), and (5) Process (clarity around the process of SUD). Table 4 presents some of the recommendations for good practice when implementing new electronic systems with the view of reusing data from that system as well as common pitfalls to avoid. These recommendations could be used by healthcare providers to help optimise SUD opportunities and to benefit from existing electronic data, 

### 4.4. Implications for Research

This review found a lack of consistency in the definitions used either explicitly or implicitly, for SUD interventions, systems, and processes in the literature and therefore sought to develop conceptual tools to facilitate a more coherent approach for research and development in the future. Clear definitions of intervention concepts and mechanisms are required if a coherent evidence base is to be built and the definition developed through this study (“the reuse of aggregated clinical or operational data from an electronic prescribing or electronic hospital pharmacy system for purposes other than direct patient care or its original purpose”) might serve as a useful starting point. Furthermore, limited detail of the SUD interventions, as well as technical and contextual barriers, creates a challenge in identifying the extent to which the results are generalizable. A model for documenting/reporting complex SUD interventions to facilitate translation into other settings is required. The steps in Figure 2 could be used to report against details of the SUD interventions to enhance the reproducibility of interventions as well as clarifying the barriers and facilitators encountered within each study setting. Future research into the effectiveness of SUD interventions is required and should include process evaluation of the factors contributing to success or failure of the intervention. The meaningful effect size and outcome measures relating to future SUD intervention studies will need to be determined based on study objectives. A mixed methods study design would be best suited for evaluating complex SUD interventions as it would consider the actual intervention being implemented, as well as relevant contextual factors affecting the outcome of the intervention.

### 4.5. Strengths and Limitations

Lack of consistent terminology in this field meant that our search strategy may not have identified all relevant studies, although efforts were made to use a comprehensive search strategy. A balance was required between sensitivity and specificity in the search terms used in order to develop a consistent definition of SUD interventions resulting in a complex search strategy. This literature review focused only on SUD from EP and EHP systems in hospitals; we excluded trigger tools, smart pumps, clinical decision support systems, and bar-code technology as well as studies in primary care. We were unable to conduct a formal quality assessment on the papers due to the heterogeneity of the literature included and diverse nature of study designs. An important limitation of the current evidence base in this area therefore concerns the methodological quality of evaluations of this type of intervention. The majority of studies were single site case studies and analyses of specific interventions using qualitative and weaker quantitative designs such as uncontrolled before and after studies. The strongest reported quasi-experimental designs for causal inference were controlled before and after studies and time series designs, though only three of the nine included interventional studies were of these types. Furthermore, in the current published research literature, there was a lack of reported detail in the processes authors followed in development of SUD interventions and any potential barriers faced in implementation, which impedes replication and generalisability of evaluation results. Improving the conceptual frameworks used to design and report SUD interventions should promote consensus on specific intervention models for evaluation using more rigorous study designs in future and consequently strengthen the evidence base.

Due to the lack of detail reported concerning the features of the feedback interventions and the perceptions of individuals receiving the data, it remains unclear as to what contributed to the success of these interventions (Table 5). There is also the possibility of publication bias in this area as studies showing little or no effect may be less likely to have been published due to institutional, publication, and individual researcher incentives not to publish negative or neutral results.

The strengths of this review are the sensitivity of our search, breadth of the included literature and of the study types included. The searches were conducted on four databases; all references of included articles were screened to identify any further relevant articles to capture a broader range of literature. During the process of the review, conceptual refinement was conducted to produce clear concise criteria for this novel topic area. To reduce the risk of bias we followed strict inclusion and exclusion criteria and a registered systematic review protocol was adhered.

## 5. Conclusions

SUD for improvement of medication safety and quality is an important strategy for modern health care organisations seeking to maximise the effectiveness of their data systems. This review describes the available evidence base for effective SUD interventions. Feedback to physicians, nurses, managers, and other staff members was the most common type of SUD intervention, with most of the included studies focusing on improving prescribing or reducing omitted doses to enhance medication safety and the majority of the data being used from EP rather than EHP systems. From synthesis of research evidence in this review an integrative framework was developed to describe the processes, mechanisms, and barriers for effective SUD. The five main factors affecting SUD that emerged from the framework were the organisation (support and data accessibility), system (data quality), users of SUD (knowledge), policies and security (data privacy), and the process of SUD (advantages and disadvantages). The methodological quality of evaluations was limited for this type of intervention; only three of the nine interventional studies used a controlled before and after studies and time series design. Further work needs to be undertaken to understand the mechanisms that maximise SUD effectiveness; greater clarity should also be provided in describing SUD interventions in order to enhance replication and generalisability of the results presented. 

## Figures and Tables

**Figure 1 pharmacy-09-00198-f001:**
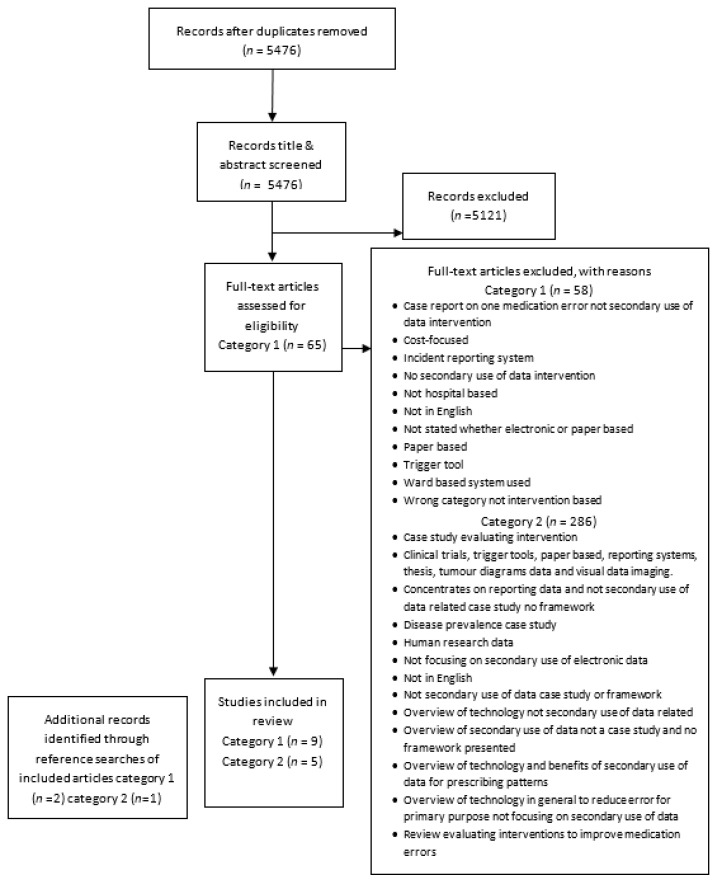
PRISMA flow diagram. Category 1: Interventional studies, Category 2: non-interventional studies.

**Figure 2 pharmacy-09-00198-f002:**
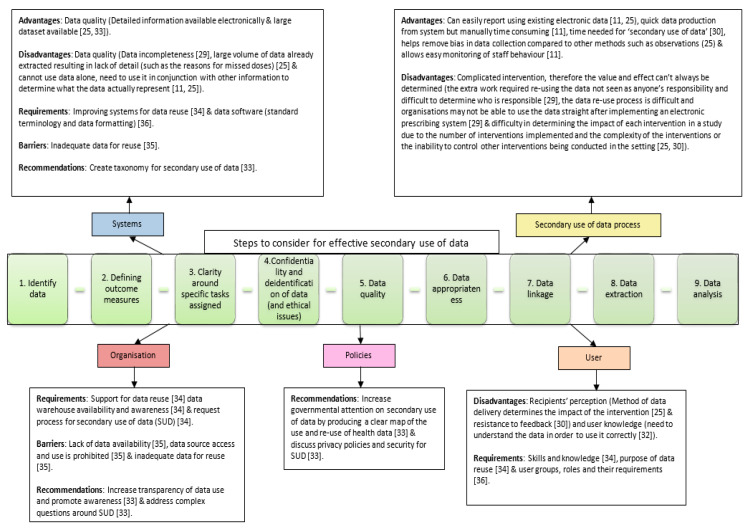
Conceptual framework around the process of secondary use of data and the potential factors influencing it—evidence base included studies.

**Table 1 pharmacy-09-00198-t001:** Search strategy adopted.

	Search Strategy 1	Search Strategy 2
Facets used	3 facets: (a) electronic databases AND (b) medication safety AND (c) [hospital or (quality or safety)]	1 facet: (Secondary$ us$ or reus$ or epurpose$) adj6 (data$ or information$ or record$)
Databases used for MeSH terms (date search conducted):	Excerpta Medical DataBASE (EMBASE) [15 August 2014], Medline [4 August 2014], and Cumulative Index to Nursing and Allied Health Literature (CINAHL) [4 August 2014].	No MeSH terms used for search strategy 2
Databases used for keywords (date search conducted):	EMBASE, Medline, IPA and CINAHL [19 March 2018].	EMBASE, Medline, IPA, and CINAHL [19 March 2018].
Search restriction criteria:	Title and abstract	Title search only
Filters applied (e.g., date, language, or publication type):	None	None

**Table 2 pharmacy-09-00198-t002:** Nine studies that explored interventions based on secondary use of data to improve the quality and safety of medication use.

Author(s) and Reference	Country and Type of Hospital Setting	Aim(s) of Study and Study Design to Evaluate Secondary Use of Data (SUD) Intervention	Secondary Use of Data Intervention(s)			
			Electronic System(s) from Which Data Were Used for SUD Intervention and Types of Data Used for the Intervention(s)	Method of Applying Data in Practice i.e., Data Are Included in Intervention and Applied in Context.	Clinical Setting and End Users	Outcome Measure(s) for Evaluation
Coleman et al., 2012 [26]	UK—type of hospital not specified	Producing upper dose limits using existing data to reduce inappropriate prescribing. Quantitative study, formula testing using retrospective data analysis	Electronic prescribing system (system not specified). 100 most frequently used drugs that were prescribed more than 100 times.	Production of upper dose limits on the electronic prescribing system.	Clinical setting not specified End user: Piloted by clinical pharmacologists	Sensitivity and specificity of the limits applied to each medication and views of piloting clinical pharmacologists. From the 28 drug form combinations available, the 86th percentile of dose gave a mean sensitivity of 95.3% and mean specificity of 97.9%.
Coleman et al., 2013 [25]	UK—Large NHS Foundation Trust	To investigate the rates of overdue doses following the implementation of electronic prescribing. Quantitative Multi-intervention study, retrospective time series analysis of weekly dose administration data.	Electronic prescribing system (Prescribing information and communications systems [PICS]) and administrative data. Overdue medication doses.	Pause function for electronic prescribing implemented, clinical dashboards produced fed back live data to end users, visual indicators and care omission meetings (for feedback) and National Patient Safety Agency rapid response.	Clinical setting: Hospital wide End users: clinical staff and managers (including board members)	Omitted doses for antibiotics and non-antibiotics. Clinical dashboards: reductions of 0.60 (95% CI = 0.26, 0.95) and 0.41 percentage points (95% CI = 0.11, 0.70) Pausing prescriptions: 0.49 percentage points (95% CI = 0.18, 0.80) and -0.28 percentage points (95% CI = −0.50, −0.07) Executive-led overdue doses root cause analysis (RCA) meetings: reductions of 0.83 (95% CI = 0.50, 1.17) and 0.97 (95% CI = 0.61, 1.32) percentage points Visual indicator for overdue doses did not result in a significant decrease in missed antibiotics or non-antibiotic doses.
Dixon-Woods et al., 2013 [11]	UK—Large NHS acute care hospital.	To improve missed doses, measure performance and overall improve quality and safety. To define the use of electronic data to generate quality and safety interventions. Qualitative Ethnographic case study	Electronic prescribing system (Hospital electronic prescribing and decision support system: ePDSS). Missed medication doses, time taken from writing antibiotic prescription to administration of first dose, venous thromboembolism risk assessment completion rates, completion of clinical observations, rates of specific infections.	The raw data from ePDSS were used with regular feedback to clinical teams (with improvement suggestions when needed via conversations, emails and meetings), dashboard displays, and support for individuals whose performance was of concern, care omission meetings, and automated emails.	Clinical setting: Two wards, the pharmacy department, and one specialist clinical unit End users: clinical teams in the areas	Qualitative interviews and opinions of staff members. Secondary use of data and feedback mechanisms not successful alone therefore RCA monthly meetings organised which were perceived to be successful.
Finnerty et al., 2002 [31]	USA—26 state psychiatric hospitals.	To use existing data to improve delivery of care for patients suffering from schizophrenia and improving adherence to guidelines. Qualitative intervention study	Electronic pharmacy database (system not specified). Daily dosage of prescribed drugs, and duration. Other demographics were obtained from administrative databases.	Use of existing electronic data to produce reports for individuals. Adherence measures confirmed using existing data and feedback from individuals. Reports presented to each clinician as feedback and illustrating deviation from recommendations by guidelines.	Clinical setting: Across all participating hospitals End users: Clinicians	Feedback from key stakeholders (clinicians) obtained. Guidelines implemented were successful; work was undertaken with end-users to ensure data are useful and met their clinical needs. Feedback from end-users was incorporated to improve data presented however there was lack of detail around how this was conducted.
Griffith and Robinson, 1996 [28]	UK—General district hospital.	To determine the scale of hypnotic prescribing and implement interventions to improve prescribing habits. Quantitative intervention study, uncontrolled before and after design.	Electronic pharmacy system (system not specified). Volume of hypnotics dispensed.	Survey sent to GP to confirm use of hypnotics upon discharge and discharge summaries examined. Electronic data and manual survey data found influenced the production of an in-house policy. Electronic data helped monitor and feedback was provided at audit meetings.	Clinical setting: Hospital (all wards with elderly patient, and 100 patients selected at random) End users: not specified (potentially clinicians prescribing)	Assessed the number of hypnotics prescribed during the first day of admission and the number of patients on hypnotics post discharge using survey for evaluation. Prescribing of temazepam reduced from 60% to 25% (*n* = 100) and ward issue reduced from 2392 to 734 tablets.
Morrison et al., 2013 [29]	UK—5 intensive care units (ICUs) across England.	To determine how 5 intensive care units (ICUs) use data effectively for direct clinical care and clinical process improvement. To analyse the methods employed by ICUs to use data effectively for clinical process improvement and direct clinical care. Qualitative Ethnographic case study.	5 Clinical information systems (Innovian (Draeger), Metavision with purchased database (iMDsoft), Metavision with own database (iMDsoft), QS (General Electric) and Carevue (Philips)). Data type not specified.	Complex intervention, no clear indication how data are being used other than for audit purposes (i.e., number of inappropriate drugs prescribed) or direct clinical care. Data quality for re-use was focused upon and initiatives used to improve this by 11 mechanisms, one of which was: using the data from the systems to improve data entry via providing positive feedback to individuals.	Clinical setting: Five ICU units End users: allied health professionals, healthcare assistant, nurses, clinical lean, consultant, local customizer, specialist registrar, senior house officer, dietician, pharmacists, and physiotherapist.	Qualitative interviews and opinions of staff members. Data had been successfully used for secondary purposes in ICUs; however, the purpose of the secondary data was not clearly stated. Difficulties experienced by the ICUs in using data for secondary purposes included: lack of primary data entry, missing information, and data quality were considered to be poor, requiring individuals to cross check information.
Nelson et al., 2005 [32]	USA—25-bed tertiary care hospital	To measure real time medication administration charting practice to improve current practice with a complex intervention to reduce error rates in administration. Quantitative intervention study, controlled before and after design.	Electronic prescribing and pharmacy system (Health Evaluation through Logical Processing: HELP system). Medication charting data and medication event reports used in conjunction. Data included: room number, patient identifier, terminal identifiers, medication name, dose, route, time of administration, computer system, charting time, reason for early or late administration, and nurse’s name.	Educational sessions on medication charting policies, error detection methods and prevention using decision support and real-time medication charting were discussed and reported back to staff. Staff then set their own target for improving real time charting. Weekly feedback was provided using real-time charting rates via posters in staff room. Poster slogans and the minutes for meetings presented to staff regularly (during morning and afternoon presentation sessions). Staff had freedom to ask any questions to investigators.	Clinical setting: Two surgical nursing units End users: all staff members	Percentage of real time charting rates calculated per day and due to awareness of the limitation of this data measure bedside charting rates were used as a second outcome measure. 8-week baseline real time charting: 59% (*n* = 16,372) for intervention unit and 53% (*n* = 18,453) for control unit. Post 12-week intervention period: 72% (*n* = 20,751) and 59% (*n* = 24,245), respectively.
Rosser et al., 2012 [27]	UK—type of hospital not specified.	To present multi-faceted interventions used to improve care and medication safety and how this can be linked to mortality. Quantitative multi-intervention case study, time series analysis.	Electronic prescribing system (PICS). Medication doses over a period of time and omitted doses from PICS and mortality rates from hospital episode statistics.	Ward based computer dashboards presenting information to clinical managers and frontline staff and monitoring of omitted medication doses. Care omission meetings led by Chief executive officer and use of electronic data for surveillance by executive team.	Clinical setting: Across hospital End users: clinical managers and frontline staff	Rate of missed doses and mortality. Overall, 16.2% reduction in mortality rates (*p* = 0.013)
Sullivan et al., 2013 [30]	USA—75-bed neonatal intensive care unit regional referral centre.	To report on development of feedback intervention and its effect on narcotic prescribing errors. Quantitative intervention-based study, uncontrolled before and after study design.	Electronic prescribing (system not specified). Use of electronic prescribing intervention data to determine errors linked to narcotic and in-house pharmacy reporting data.	Personalised and generalised email feedback provided to prescribers and prescriber feedback taken on board to improve systems (feedback strategy and bidirectional communication between prescriber and feedback team).	Clinical setting: Neonatal intensive care unit End users: nurse practitioners, physician assistants, and clinicians	Pharmacy interception rates on narcotics errors (reduced by 83%), number of days between errors (3.94 to 22.63 days), antibiotic errors unaffected by intervention (remained at 2.14 days) and read report function helped determine reading rates on prescriptions.

**Table 4 pharmacy-09-00198-t004:** Recommendations for good practice and common pitfalls to avoid when implementing electronic systems to optimise secondary use of data.

Recommendations for Best Practice	Avoid These Common Pitfalls
Discuss with existing organisations with similar electronic systems to consider their experiences with data reuse, if any.	Discussing data reuse opportunities only after system implementation, reducing the possibility of optimizing data reuse
Engage with staff and review what data your organisation would benefit from that could be used for secondary purposes, e.g., for CQUIN targets, improving prescribing habits, and meeting national standards set	Focusing solely on the primary purpose of the electronic system
Have a data warehouse (plus any additional hardware and software required)	Copying across existing reports from old system; additional opportunities exist when implementing a new system to improve the reporting functionality
Produce a policy outlining the data privacy issues and guidelines on secondary uses of data without compromising data privacy	Assuming data is always correct, without understanding the documentation processes involved with the data presented
Produce secondary use of data taxonomies, and standardize terminology and data formatting	Accepting the data at face value without understanding how data are being captured and its reflection of the process/tasks performed
Inform others in your organisation that secondary use of data opportunities exist and will be supported	
Implement an easier process for staff to review what data are already captured in your organization and enhance data accessibility	
Have a better system reporting functionality (determine the data timeliness, accuracy, validity, and completeness)	
Include data quality reports to improve data entry and enhance data transparency	
Ensure data anonymisation	
Have easier data extraction methods when implementing a new system	
Understand the caveats associated with the different data entries (i.e., knowledge of actual system usage resulting in the data produced rather than how data should be entered in theory)	
Have designated individuals who have clear knowledge around the strengths and weaknesses of the data collected	
Educate and train staff so they have the right skillset and knowledge relating to the data	
Engage recipients of data in order to maximise the SUD intervention impact and have a positive outcome	

**Table 5 pharmacy-09-00198-t005:** Interventional studies that combined secondary use of data with feedback and characteristics of each feedback intervention used.

Authors	Characteristics of Feedback That Make It Successful [1]						
	Data Are Valid Based on the Receivers Perspective	Credibility of Data in Organisation (for Receivers to Believe the Data Presented)	Source and Timeliness of Data	Benchmarking (Comparing Results amongst Others and Promoting Healthy Competition)	Leaders Presenting Data (Trusted Individuals Presenting Data)	Data Are Individualised (Personalised Data)	Constant Feedback (Not a One-off Account of Feedback)
Coleman et al., 2013 [25]	Not stated in paper.	Not stated.	Clinical dashboard available for staff. Reports produced feeding back data and care omission meetings constructed for feedback.	Yes, against ‘acceptable rates’ set within the hospital.	Care omission meetings lead by executive members.	Directorate level emails not personalised data.	Not clear if dashboard was live (regularly accessible to user), weekly emails and monthly meetings.
Dixon-Woods et al., 2013 [11]	Yes data validity not questioned.	Not stated.	The raw data from ePDSS were used with regular feedback to clinical teams (with improvement suggestions when needed via conversations, emails and meetings), dashboard displays, and support for individuals whose performance was of concern, care omission meetings, and automated emails.	Against set standards within the organisation.	Executive team leading meetings.	Yes and support provided for individuals causing concern.	Monthly meetings and dashboard displays available for users.
Finnerty, M., et al., 2002 [31]	Receivers had opportunity to feedback on report, but nothing stated around data validity.	Receivers knew data source.	Use of existing electronic data to produce reports for individuals and monthly data collected.	Compared against guidance.	Reports produced and piloted not clear if presented by leaders.	Use of existing electronic data to produce reports for individuals. Reports presented to each clinician as feedback and illustrating deviation from recommendations by guidelines.	Input for feedback report production done twice.
Morrison et al., 2013 [29]	Not stated.	Not stated.	using the data from the systems to improve data entry via providing positive feedback to individuals.	Yes, positive results stated.	Senior staff.	Yes, by stating people adhering to correct data entry.	Not stated.
Nelson et al., 2005 [32]	Not stated.	Staff had freedom to ask any questions to investigators.	Real-time medication charting was discussed and reported back to staff. Poster slogans and the minutes for meetings presented to staff regularly (during morning and afternoon presentation sessions).	Target for improving real time charting was set by senior staff.	Not stated.	Not stated.	Weekly feedback was provided using real-time charting rates via posters in staff room.
Rosser et al., 2012 [27]	Not stated.	Not stated.	Ward-based computer dashboards presenting information to clinical managers and frontline staff and monitoring of omitted medication doses.	Not stated.	Care omission meetings led by Chief executive officer and use of electronic data for surveillance by executive team.	Not stated.	Clinical dashboard available to staff.
Sullivan et al., 2013 [30]	Not stated.	Receivers aware of feedback programme as involved in the piloting.	Dashboard using electronic data and data collected every two weeks.	General and personalised feedback available not clear statement around benchmarking performance.	Presented via email and by pharmacists.	Personalised and generalised email feedback provided to prescribers and prescriber feedback taken on board to improve systems (feedback strategy and bidirectional communication between prescriber and feedback team).	Yes, every two weeks.
Griffith and Robinson, 1996 [28]	Note stated.	Not stated.	Electronic pharmacy data and survey data.	Not stated.	Not stated.	Not stated.	Not stated.

## Data Availability

No new data were created or analysed in this study. Data sharing is not applicable to this article.

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
