# Peer review of "The Secondary Use of Data to Support Medication Safety in the Hospital Setting: A Systematic Review and Narrative Synthesis"

_pharmacy, 2021, doi:10.3390/pharmacy9040198_

Round 1

Reviewer 1 Report

This paper is very aptly summarized. However, the table does not conform to the format. Therefore, minor modifications are required.

This paper is also very useful for readers. The majority of the SUD interventions were successful in improving medication safety, however what contributes to this success needs further exploration. I look forward to future research by the authors.

Author Response

Thank you for the helpful feedback and comment regarding the table formatting and value of the paper for readers.

The review findings do indeed support the case for SUD interventions in improving medication safety and we have drawn out/discussed the characteristics of successful SUD interventions where possible.  However, one of the findings from the review is that the factors contributing to the success of the interventions were not clear in the primary literature (lines: 497-499) and we have therefore suggested future studies use the SUD framework produced to allow transparency on the factors contributing to the success of the interventions (lines 501-503).  In response to the reviewer’s comment, we have now added in the “implications for research” section the following text “Future research into the effectiveness of SUD interventions is required and should include process evaluation of the factors contributing to success or failure of the intervention” (lines 503-505).

We believe the reviewer was referring to table 5 in their comment; table 5 has now been amended (text has been centred, text font style changed to Palantino Linotype. 

Reviewer 2 Report

This paper attempts to describe, structure and generalize the methods for evaluating data from electronic prescribing. This in itself is a very difficult problem and the difficulties begin with the definitions and the selection of literature. The authors have taken great care to find the appropriate papers according to their definition. It could well be argued that more papers could have been included which also have contributed to establish electronic prescribing as a method leading to better drug safety. What benefit for the healthcare provider may result as mentioned in the introduction remains unclear, but raises the ethical question of data privacy. The evaluation of the methods for making the data useful in patient management highlight the limitations of this ardeous endeavor and point to the need for better decision support systems. The authors should  have discussed the relation beetween efforts spent and the vey limited results obtained so far by this method.

Author Response

Thank you to the reviewer for their constructive comments. We have addressed the comments as follows:

Comment: What benefit for the healthcare provider may result as mentioned in the introduction remains unclear, but raises the ethical question of data privacy.

We had to strike a balance between sensitivity and specificity in the search algorithm in order to develop a consistent definition of SUD interventions for evidence synthesis.  We did this by focusing on the conceptual definition. We have now added in the following sentence under strengths and limitations to address this point: “A balance was required between sensitivity and specificity in the search terms used in order to develop a consistent definition of SUD interventions resulting in a complex search strategy.” Lines 513-515. The outcomes achieved successfully by the original research papers included in this review are the benefits healthcare providers could achieve in practice. The implications for practice section 4.3 in the original manuscript addresses the areas that individuals may need to review when trying to reuse electronic data. We have added in the ‘recommendations for best practice’ in table 4 based on the this and the third reviewer’s comment below and have added in the following sentence: “These recommendations could be used by healthcare providers to help optimise SUD opportunities in organisations to benefit from existing electronic data” lines 484-485.

We have also stated the following to elaborate on our view of using the SUD framework to optimise reuse of existing data “Therefore healthcare providers could review these factors in their organisations before reusing existing electronic data, to help optimize the outcomes they wish to achieve and reap the benefits of reusing existing data.” (Lines 454-456).

In the original paper submitted the following sentence aimed to encourage future work around data privacy as we felt the included literature did not cover all aspects: ‘Future developments need to address concerns reported in the empirical literature concerning policies relating to privacy and security of data for secondary use’ (lines 466-468) – this sentence has been amended slightly to ensure the message is relayed clearly. Similarly, ‘policy’ was included in the original paper submitted as part of the SUD framework that should be considered by future SUD users. ‘Policy (the security and data protection need to be considered when reusing data i.e. potential risk of re-identification)’ (lines 479-481).

Comment: The authors should have discussed the relation between efforts spent and the very limited results obtained so far by this method:  It can be difficult to obtain data from electronic systems to utilise it for secondary purposes, and our work established the current evidence base for SUD interventions but additionally highlights the challenges in making effective secondary use of data. However, in light of the reviewers comment we have added the following “Further research and development is required to realise the potential of routinely available data for medication safety, increase the return on investment by healthcare providers in making secondary use of data and including better linkage to systems that can facilitate and optimise secondary use, such as decision support systems.” (lines 468-471). In the original paper that was submitted, we have stated the limitations experienced by some of the primary studies, where mentioned in the published paper. For example some studies experienced practical limitations such as data volumes, this point was addressed in the following sentence included in the original paper “Practical limitations were experienced in one study where large datasets could not be analysed; this could have provided an insight into organisational factors affecting the study outcome” (lines 347-349). 

Reviewer 3 Report

This manuscript provides a very detailed analysis of available literature related to secondary use of data from electronic prescribing and electronic hospital pharmacy systems in the hospital context. The systematic review procedure is elaborately documented, but tends to detract from the most important message of the manuscript. In this reviewers’ opinion the article can have more impact if more emphasis is given to the practical implications for the development of electronic systems, which can be suitable for SUD, and their use in practice. A table, summarizing do’s and don’ts, can be added to section 4.3.

The use of the terminology ‘advantages’ and ‘disadvantages’ of SUD (line 305 and fig. 2) is prone to misunderstanding. It suggests that SUD as such can be advantageous or disadvantageous, irrespective of the quality of implementation of systems and use of available data. However, in this reviewers’ opinion the important issue is whether the potential advantages of SUD can be realized by the quality of implementation, data input and data use. It would be better to rephrase advantages and disadvantages in terms of opportunities and threats (or risks) and formulate criteria for good practice.

I wonder whether the appendices add anything useful for the general reader. These appendices probably are direct copies of instructions, used during the literature search process, but are a bit over-detailed (as referred to above). If they are kept in the final manuscript, search query 68 should be corrected and table 2 of Appendix A can be deleted. Readability of Appendix B can be improved by starting with the figures and explaining each step in the decision tree in the text subsequently. What is the difference between Table 3 and Appendix D?

A few syntactic and layout problems are detected in the manuscript:

  1. Layout of the tables is inconsistent (position of heading, boxed fields).
  2. Fig. 2 has a double caption.
  3. Table 1 is broken over the page border.
  4. Position of Figure 1 and white spaces around needs to be improved.
  5. Line 466: “modern health care organisations seeking to maximise their investment in information and data systems”. Hopefully the health care organisations are not seeking to maximise their investment in the systems, but are concentrating on the effectiveness of those systems.

Author Response

We have addressed the reviewer’s comment regarding the practical implications for the development of electronic systems and have merged this with the reviewer’s second recommendation below resulting in the addition of recommendations for best practice and pitfalls to avoid (changes presented in table 4).

The terms advantages and disadvantages were not changed as the terms reflect the pros and cons of reusing existing data as reflected in the included studies. ’Opportunities and threats’ are conceptually different from advantages and disadvantages. Opportunities are more provisional whereas advantages are inherent in nature and grounded in real experience. However, we have now framed the advantages and disadvantages as recommendations for best practice and pitfalls to avoid in table 4 as per the reviewer’s recommendation. 

Appendix D was a more detailed version of table 3 but in light of the comment made regarding duplication of material, we have removed the appendices from the resubmission pending further editorial guidance. The references to all the appendices have been removed from the main text except for appendix 1 which we have kept to comply with the PRISMA guidelines to include details of the search strategy.

Thank you for identifying these issues:

1.      The tables have now been consistently formatted.

2.      The double caption in figure 2 has been removed

3.      Table 1 has now been repositioned to fit on 1 page

4.      Figure 1 has been amended for clarity

5.      The sentence has now been changed to: “SUD for improvement of medication safety and quality is an important strategy for modern health care organisations seeking to maximise the effectiveness of their data systems.” Line 549-551